# Application of Exogenous Protectants Mitigates Salt-Induced Na^+^ Toxicity and Sustains Cotton (*Gossypium hirsutum* L.) Seedling Growth: Comparison of Glycine Betaine and Salicylic Acid

**DOI:** 10.3390/plants10020380

**Published:** 2021-02-17

**Authors:** Abdoul Kader Mounkaila Hamani, Jinsai Chen, Mukesh Kumar Soothar, Guangshuai Wang, Xiaojun Shen, Yang Gao, Ranjian Qiu

**Affiliations:** 1Farmland Irrigation Research Institute, Chinese Academy of Agriculture Sciences/Key Laboratory of Crop Water Use and Regulation, Ministry of Agriculture and Rural Affairs, Xinxiang 453002, China; m_abdoulkader@yahoo.com (A.K.M.H.); 82101182107@caas.cn (J.C.); mukeshksootar@gmail.com (M.K.S.); wangguangshuai@caas.cn (G.W.); shenxiaojun8003@163.com (X.S.); 2Graduate School of Chinese Academy of Agricultural Sciences, Beijing 100081, China; 3Collaborative Innovation Center on Forecast and Evaluation of Meteorological Disasters, Jiangsu Key Laboratory of Agricultural Meteorology, Nanjing University of Information Science and Technology, Nanjing 210044, China

**Keywords:** biomass accumulation, cotton, glycine betaine, growth, ions, salicylic acid, soil salinization

## Abstract

Soil salinization adversely affects agricultural productivity. Mitigating the adverse effects of salinity represents a current major challenge for agricultural researchers worldwide. The effects of exogenously applied glycine betaine (GB) and salicylic acid (SA) on mitigating sodium toxicity and improving the growth of cotton seedlings subjected to salt stress remain unclear. The treatments in a phytotron included a control (CK, exogenously untreated, non-saline), two NaCl conditions (0 and 150 mM), four exogenous GB concentrations (0, 2.5, 5.0, and 7.5 mM), and four exogenous SA concentrations (0, 1.0, 1.5, and 2.0 mM). The shoot and roots exposed to 150 mM NaCl without supplementation had significantly higher Na^+^ and reduced K^+^, Ca^2+^, and Mg^2+^ contents, along with lowered biomass, compared with those of CK. Under NaCl stress, exogenous GB and SA at all concentrations substantially inversed these trends by improving ion uptake regulation and biomass accumulation compared with NaCl stress alone. Supplementation with 5.0 mM GB and with 1.0 mM SA under NaCl stress were the most effective conditions for mitigating Na^+^ toxicity and enhancing biomass accumulation. NaCl stress had a negative effect on plant growth parameters, including plant height, leaf area, leaf water potential, and total nitrogen (N) in the shoot and roots, which were improved by supplementation with 5.0 mM GB or 1.0 mM SA. Supplementation with 5.0 mM exogenous GB was more effective in controlling the percentage loss of conductivity (PLC) under NaCl stress.

## 1. Introduction

Soil salinization is an increasing problem in agriculture, affecting an area of ~800 million ha that accounts for more than 7% of the total land area worldwide [1]. Salt accumulation in the soil is often caused by irrigation with water containing sodium chloride (NaCl) [2]. NaCl-associated salinity has recently become a focus in abiotic stress research on non-halophytic plants. NaCl stress-related effects, such as ion toxicity and low water potential, lead to numerous changes in plant metabolism [3].

Both Na^+^ and Cl^−^ ions contribute to soil salinization. The detrimental impact of salinity is often associated with excessive uptake of Na^+^ or Cl^−^ leading to nutritional imbalance. High Na^+^ levels in the rhizosphere cause Na^+^ uptake by root cells via K^+^ transporters. One of the most harmful effects of salinity stress on the plant is the disturbance of ion homeostasis [4]. Plants can adopt several strategies to avoid an adverse decrease of the K^+^/Na^+^ ratio by reducing Na^+^ entry into the cells and by removing Na^+^ from the cells, along with Na^+^ compartmentalization into the vacuole to avoid disrupting any cellular function [5]. Thus, plants respond to salinity stress by reducing Na^+^ ion uptake and its translocation from the roots to shoots. One critical physiological response to salinity stress in plants is the increasing accumulation of organic or inorganic solutes to reduce the tissue osmotic potential [6]. Glycine betaine (GB) is a quaternary ammonium compound that has an essential function in cellular osmoregulation in plants under abiotic stress [7,8]. Exogenous foliar supplementation with GB has been suggested for improving salinity stress resistance in plants due to its role in Na^+^/K^+^ discrimination and ion homeostasis under saline conditions [9]. Previous studies reported that a low dose of exogenous GB is very effective in decreasing K^+^ loss in Arabidopsis (*Arabidopsis thaliana*) and barley (*Hordeum vulgare*) [10]. Low concentrations of exogenous GB also sustained a higher K^+^ content in rice (*Oryza sativa*) seedlings under salt stress conditions [11]. Salicylic acid (SA) is a vital phytohormone functioning as a signaling molecule in plant growth and modulating various salt stress responses [12]. Karlidag, et al. [13] reported that foliar supplementation with SA alleviates the adverse effects of salinity stress by increasing the N, P, K^+^, Ca^2+^, and Mg^2+^ contents and by decreasing the Na^+^ content in strawberry roots and leaves.

Plant growth under salt stress is often impaired by the limited capacity of roots for water uptake from the rhizosphere and transfer to the shoot because of the osmotic effect of soil salinity. The nutrient amount moved to the shoot depends on the water volume translocated from roots to shoot [14]. Thus, high salinity decreases plant growth by impairing the plant’s osmotic or ion homeostasis. Saline conditions also negatively affect plant height, primary and lateral root growth, leaf augmentation, stem thickness, and root and shoot biomass. Cotton, as the most abundant fiber source for textiles worldwide, is classified as a medium salt-tolerant crop that can tolerate a salt threshold of 7.7 dS m^−1^. However, its growth and development are still substantially decreased under high saline conditions, especially at the seedling stage [15]. The negative impact of salinity is exacerbated in cotton plants exposed to saline conditions for prolonged periods [16]. High salinity often induces a wide range of metabolic processes. For instance, salt stress decreased the total nitrogen (N) content in the root and stem of *Sesbania* sp. [17]. However, exogenously sprayed GB and SA had a positive effect on crop growth and yield under saline conditions [2]. Typically, a wide array of physiological processes in plants subjected to a saline regime, including growth and development, is known to be affected by foliar supplementation with exogenous SA. Liu, et al. [18] demonstrated that exogenously sprayed SA might improve plant resilience to salinity by functioning as a growth regulator.

To date, the effects of foliar-sprayed GB and SA on NaCl-stressed cotton seedlings are not well understood. The main goal of our study was to investigate the protective effects of exogenously sprayed GB and SA on cotton seedlings grown under a high-salinity regime (150 mM NaCl) by monitoring and analyzing the response of growth and biomass accumulation, percentage loss conductivity (PLC), and ion (Na^+^, K^+^, Ca^2+^, and Mg^2+^) content.

## 2. Results

### 2.1. Effect of Exogenous GB and SA on Cotton Seedling Growth and Nitrogen Accumulation

The growth parameters plant height, leaf area, and leaf water potential (LWP) of 150 mM NaCl-stressed cotton seedlings without supplementation (SS) were significantly reduced by 20%, 35%, and 80%, respectively, compared with those of the control seedlings (CK) (Figure 1). However, exogenous foliar supplementation with GB and SA improved these parameters compared to the SS treatment. Supplementation of NaCl-stressed seedlings with 2.5, 5.0, and 7.5 mM GB or 1.0 and 1.5 mM SA significantly increased their plant height compared with SS-treated seedlings (Figure 1a). Moreover, the NaCl-stressed seedlings treated with 2.5 and 5.0 mM GB or 1.0 mM SA had a significantly larger leaf area than the SS-treated seedlings (Figure 1b). All concentrations of both foliar treatments —GB and SA—significantly increased LWP in NaCl-stressed cotton seedlings compared with SS-treated seedlings (Figure 1c).

Results of foliar-applied GB and SA on biomass accumulation in cotton seedlings grown under 150 mM NaCl stress are presented in Figure 2. The shoot and root biomasses were significantly reduced by 37% and 35%, respectively, in SS-treated seedlings, compared to CK seedlings. Only NaCl-stressed seedlings treated with foliar spray containing 5.0 mM GB had higher shoot and root biomasses than SS-treated seedlings.

The total nitrogen (N) content in the shoot and roots of SS-treated cotton seedlings was significantly decreased by 30% and 21%, respectively, compared with that in CK seedlings. The total N content in the shoot and roots of all NaCl-stressed cotton seedlings treated with either foliar spray at any concentration was significantly higher than in SS-treated seedlings (Figure 3). The increase in the total N content in the shoot and roots was similar with either foliar treatment—exogenous GB or SA—in NaCl-stressed seedlings. The maximal accumulation of total N in the shoots and roots was measured in seedlings treated with 1.0 mM SA.

### 2.2. Effect of Exogenous GB and SA on the PLC

Across all the experimental treatments, the percentage loss of conductivity (PLC) progressively decreased with increasing distance from the soil surface to the cut stem. It became constant at 25 cm from the soil surface to the cut stem, corresponding to the native state of embolism under the general experimental conditions. At 5 cm from the soil surface to the cut stem, the PLC value was higher under SS treatment (close to 100%) and lower under CK conditions (less than 80%) (Figure 4). From 5 to 30 cm from the soil surface to the cut stem, the exogenous foliar applications of GB and SA, especially 5.0 mM GB and 1.0 mM SA, significantly reduced the PLC value under NaCl stress near to that under CK conditions.

### 2.3. Effect of Exogenous GB and SA on Ion Concentrations in the Shoot and Roots of Seedlings

Table 1 and Table 2 present the impact of exogenous GB and SA on the ion concentrations in cotton seedling shoot and roots under 150 mM NaCl stress. In the shoot and roots under NaCl, the levels of K^+^, Ca^2+^, and Mg^2+^, along with the K^+^/Na^+^ ratio, were significantly lower than those in the well-watered seedlings. In contrast, the Na^+^ levels in the shoot and roots were significantly higher in SS-treated seedlings than in CK seedlings. All exogenous GB and SA concentrations significantly increased the K^+^, Ca^2+^, and Mg^2+^ content, along with the K^+^/Na^+^ ratio, in the shoot and roots of cotton seedlings under the NaCl regime, while the same treatments significantly reduced the Na+ content in the shoot and roots under salt stress. However, the Ca^2+^/Mg^2+^ ratio in seedlings was not significantly affected by NaCl in the shoot but was significantly reduced in the roots compared with this ratio in CK seedlings. Moreover, exogenous foliar supplementation with either GB or SA did not significantly change the Ca^2+^/Mg^2+^ ratio in the seedling shoot under the NaCl regime (Table 1). In contrast, the Ca^2+^/Mg^2+^ ratio in the roots was significantly decreased under all exogenous GB and SA concentrations in NaCl-stressed seedlings (Table 2).

Among all exogenous foliar GB and SA treatments, the medium dose (5.0 mM) of GB was optimal in mitigating Na^+^ uptake by the roots and its accumulation in the shoot of NaCl-stressed seedlings. Significant negative linear relationships were determined between the shoot or root Na^+^ content and the shoot or root biomass accumulation, as shown in Figure 5.

### 2.4. Relationship between PLC and Shoot Nutrients

The 150 mM NaCl condition without exogenous foliar treatment had the highest PLC values at the cotton stem cut at different distances from the soil surface. However, foliar treatment with 5.0 mM GB or 1.0 mM SA led to a significant reduction in PLC, most prominently at the stem cut 30 cm from the soil surface. Our results indicated that the cotton PLC 30 cm from the soil surface to the stem cut was significantly correlated with the shoot nutrient content (Figure 6). The PLC was significantly negatively correlated with the shoot total N content, shoot Ca^2+^ content, and shoot K^+^/Na^+^ ratio but significantly positively correlated with the shoot Na^+^ concentration.

## 3. Discussion

Soil salinization is a major environmental stress that increases ionic toxicity and osmotic stress in plants, resulting in decreased plant growth and functions. The accumulation of toxic ions was reported to be the specific effects of salt stress on plant metabolism [19]. The mechanisms of adaptation of plants to salt-induced ionic and osmotic stress are associated with GB-mediated osmotic regulation [20]. SA is known to induce the salt tolerance of plants by decreasing osmotic and ionic stress [19].

The current investigation suggests that exogenous foliar-applied GB or SA may help to protect cotton seedlings against the adverse effects of salinity stress. Several studies reported positive effects of exogenously applied GB on plants such as *Oryza sativa* and *Lycopersicon esculentum* under saline conditions [7,11,21]. It was reported that exogenous application of SA reduced the Na uptake of plants and/or increased the uptake of N, P, K, Ca, Mg, and other minerals compared to the control treatment under salt-stress conditions.

Saline conditions can negatively affect the nutrient content of plants, causing plant exposure to osmotic stress and affecting the content of specific nutrients. Saline exposure can also lead to a reduction in yield and crop performance as a result of nutritional imbalance [22]. The ability of plants to reduce ion influx into the cytoplasm is of great importance to salinity tolerance [19]. In this study, exogenous foliar supplementation with GB and SA was effective in enhancing nutrient uptake by cotton seedlings cultured under 150 mM NaCl stress. Both GB and SA contributed to a reduction of the Na^+^ content in the shoot and roots of salt-exposed cotton seedlings (Table 1 and Table 2). El-Tayeb [23] similarly reported that exogenous foliar supplementation with SA reduced the shoot and root Na^+^ content and improved the K^+^, Ca^2+^, and P levels in all components of barley seedlings grown under a saline regime compared with well-watered plants. Gunes, et al. [24] and Yildirim, et al. [25] also demonstrated that exogenously sprayed SA inhibited Na^+^ uptake and increased N, P, K, and Mg^2+^ accumulation in maize plants subjected to salinity stress compared with plants grown under normal conditions. However, it was demonstrated that under a saline regime, foliar-sprayed GB improved root nutrient uptake and lowered the Na+ content in plants [13,26,27]. Similarly, Gadallah [28] reported that foliar supplementation with GB caused an important reduction in the leaf Na^+^ content and a significant improvement in the accumulation of nutrient elements in the leaves of plants subjected to salinity stress. These findings resemble our results on ion responses to exogenous GB and SA under the NaCl regime (Table 1 and Table 2). Yildirim, et al. [5] suggested that the plant growth improvement induced by exogenously sprayed GB could enhance the accumulation of K^+^ and Ca^2+^ in plant organs, sustain an elevated K^+^/Na^+^ ratio, and reduce the leaf Na^+^ content. In our study on cotton seedlings grown under the 150 mM NaCl regime, both exogenous GB and SA significantly increased the nutrient element uptake by the roots and significantly lowered the Na+ content in the shoot and roots.

It is critical to discuss the impact of foliar-sprayed GB and SA in maintaining the growth characteristics for enhancing crop productivity under saline conditions. Across all tests performed on the seedlings, the NaCl regime without supplementation significantly affected the growth parameters, including plant height, leaf area, and LWP (Figure 1). Growth and development were delayed in cotton seedling under the 150 mM NaCl regime when compared to the status of well-watered control seedlings. The current study showed that exogenous foliar supplementation with GB or SA is a suitable approach for maintaining growth in cotton seedlings under NaCl stress. Results showed that the medium concentration (5.0 mM) of exogenous GB and the lowest concentration (1.0 mM) of exogenous SA had the strongest effect in enhancing the cotton seedling growth parameters under the NaCl regime. A reduction in plant height and dry mass was observed in cotton plants subjected to salinity stress in an earlier study by El-Beltagi, et al. [29]. In contrast to our findings, Heuer [30] found that exogenously sprayed 5.0 mM of GB decreased LWP, as well as fresh and dry biomasses under a saline regime, which could be attributed to the mode of application or the choice of the actual dose. Plant growth reduction under saline conditions can reportedly result in ion-specific toxicity during ion uptake under NaCl stress or osmotic damage [10]. Hence, we hypothesized that the reduction of the cotton seedling leaf area under the 150 mM NaCl regime might be due to a decrease in turgor pressure and a blockage in cell division and multiplication. A similar suggestion was made by Ahmed, et al. [31], who assessed the underlying mechanisms for leaf area reduction under saline regimes. SA plays a key function in adjusting plant growth and development under numerous environmental stress conditions [32]. However, it is still unclear whether exogenously sprayed SA can alleviate the adverse effects of salinity [33]. Exogenous foliar supplementation with SA is expected to control stomatal opening under stress conditions, which reduces the water loss by transpiration and helps the plants to perform gas exchange, sustain turgor, and, eventually control plant productivity under stress conditions [34]. Kurepin, et al. [8] reported that exogenously sprayed GB positively affected plant growth characteristics under abiotic stress conditions. In this study, the exogenous foliar application of GB on cotton seedling leaves significantly improved the growth parameters of the seedlings under the NaCl regime (Figure 1). Among the different experimental treatments, the medium concentration (5.0 mM) of GB and the lowest concentration (1.0 mM) of SA increased the cotton seedling growth parameters under the NaCl regime compared with those under NaCl stress alone.

Plant growth reductions, such as the decline of the shoot and root biomasses, are among the major consequences of the saline regime; this often diminishes the yield in most plant species [2]. These findings are consistent with our results (Figure 2), where NaCl stress alone considerably reduced the shoot and root biomasses compared with the normal growing conditions. In this study, exogenously sprayed GB and SA were chosen as a preventive treatment to alleviate plant growth reduction under the saline regime. Both foliar-applied treatments—GB and SA—mitigated the NaCl-induced growth inhibition and led to increased biomasses of the shoot and roots compared with the NaCl-stressed treatment without supplementation. Our data agree with earlier reports by El-Tayeb [23] and Arfan, Athar and Ashraf [33] on exogenously sprayed SA that alleviated the adverse effects of saline stress on wheat and barley growth and development. However, it has been well established that exogenous foliar supplementation with GB mitigates the effects of salinity on root and shoot expansion in many plant species, including eggplant [26], okra [27], and lettuce [5]. These observations were supported by Abbas, et al. [26], who demonstrated that under a saline regime, tomato and eggplant growth and yield were positively affected by exogenously sprayed GB. As expected, NaCl stress caused a significant Na+ accumulation in the shoot and roots of cotton seedlings, which was associated with a reduction in the biomass of both the shoot and roots. However, when sprayed on the leaves, the medium dose (5.0 mM) of GB significantly increased the biomass in the shoot and roots under NaCl stress compared with the seedlings under NaCl stress without supplementation. Our results also indicated some significantly negative relationships between the shoot or root Na+ content and the shoot or root biomass (Figure 5).

The effects of exogenous GB and SA on growth parameters such as the total N content and PLC in cotton seedlings subjected to NaCl stress are critical observations. In our experiment, the treatment with 150 mM NaCl alone caused a decrease in the total N content in the shoot and roots compared with the treatment control. In contrast, all treatments with exogenous GB and SA significantly improved the total N content in the shoot and roots compared with the effect of the NaCl stress treatment without supplementation (Figure 3). Improving the N content is essential for maintaining the vegetative growth in plants under stress conditions. Our results agree with those obtained by Youssef, et al. [35], who showed that exogenous foliar supplementation with GB or SA improved the N content in cucumber leaves under saline conditions. Similarly, Yildirim, et al. [25] also found that exogenously sprayed SA enhanced the accumulation of N in the leaves and roots of cucumber plants subjected to salinity stress. Our data showed that the PLC values were significantly higher in NaCl-stressed seedlings than in well-watered seedlings, similar to the impact on plants affected by drought stress. Starting from the base of the stem to the upper part of the plant, the PLC value decreased, i.e., the PLC was inversely correlated with the distance from the soil surface, indicating that hydraulic conductivity improved from the base to the upper part of the stem. Our findings are consistent with those obtained by Rajput, et al. [36], who showed that the PLC in *Populus euphratica* was significantly increased by 52% under 150 mM NaCl stress. A salt-induced increase in PLC might result in a decrease in xylem hydraulic conductivity, leading to an abnormal transfer of nutrients from the roots to the shoot. A previous study reported that high salinity caused a general reduction in xylem hydraulic conductivity [36]. As predicted, exogenously sprayed GB and SA positively affected the PLC value and nutrient (Na^+^, Ca^2+^, and K^+^) content in NaCl-stressed cotton seedlings. A regression analysis revealed significant linear relationships between PLC and the nutrient (Na^+^, Ca^2+^, and K^+^/Na^+^) content (Figure 6).

In a recent study, we observed that as a response to 150 mM NaCl stress, the endogenous biosynthesis of GB and SA was significantly increased compared with the well-watered treatment control [37]. Our findings are in good agreement with those of Chen and Murata [38,39], who reported that plants synthesize GB under environmental stress, including salinity. Similarly, Pál, et al. [40] reported that SA is endogenously accumulated to protect plants against the negative impacts of abiotic stresses, including salinity and many others, and ensuring plant growth and development. However, despite a significant accumulation of GB and SA in cotton seedling leaves under NaCl stress, the seedlings were unable to mitigate the adverse effects caused by NaCl stress [37]. In a previous study, we found exogenous foliar-applied GB significantly enhanced the endogenous accumulation of GB but had a non-significant effect on the accumulation of endogenous SA compared with changes under NaCl stress without supplementation. In contrast, the SA-containing foliar spray significantly stimulated both endogenous GB and SA accumulation [37]. In agreement with our observations, exogenously sprayed SA ameliorated the accumulation of endogenous GB in *Vigna radiata* subjected to salinity stress [19]. However, the endogenous GB content could be increased by exogenous GB applied to *Lolium perenne* leaves and stems [41] and *Triticum aestivum* [42] under saline conditions. It may be concluded that exogenous foliar supplementation with SA alleviates adverse effects of salt stress by increasing the synthesis of GB. The effects of SA on GB synthesis alleviated the negative effects of salt stress on plant growth.

## 4. Materials and Methods

### 4.1. Experimental Material

The experiment was conducted in a phytotron at the experimental station of the Institute of Farmland Irrigation Research, Chinese Academy of Agricultural Sciences (35°08′ N, 113°45′ E, and 80.77 m altitude), located in Xinxiang city in Hanan Province, China. Environmental controls of the phytotron included a day/night temperature of 30/20 °C, a relative humidity of 50%, and a 14-h photoperiod with active photosynthetically radiation of 350 µmol m^−2^ s^−1^ supplied by Light-Emiting Diode (LED) lamps from 6:00 to 20:00. The cotton (*Gossypium hirsutum* L.) variety Xinluzhong-37, a leading variety in the salty soil area of Southern Xinjiang, was used in this study as experimental material. Seeds were purchased from Tahe (Seed CO., LTD), Alaer city in Xinjiang Province. After disinfection for 30 min in 3% hydrogen peroxide, cotton seeds were rinsed several times with deionized water to remove any potential contamination. Seeds were germinated in a nursery for one week, and then the seedlings were transplanted in plastic pots with one plant per pot. The diameter and height of each pot were 16 cm and 18 cm, respectively. Dry sandy sterile soil, weighing 2.5 kg, was added to each pot.

The cotton seedlings were regularly watered with half-strength Hoagland’s solution for 20 days from the day of transplanting to ensure their growth in the sterile sand. The nutrient solution composition in g × L^−1^ was as follows: 236.2 Ca(NO_3_)_2_•4H_2_O, 101.1 KNO_3−_, 40 NH_4_NO_3−_, 61.6 MgSO_4_•7H_2_O, 34 KH_2_PO_4_, 18.6 KCL, 3.671 Fe-EDTA and microelements (1.546 H_3_BO_3−_, 0.396 MnCl_2_•4H_2_O, 0.575 ZnSO_4_•7H2O, 0.125 CuSO_4_•5H_2_O, 0.036 CoCl_2_•6H_2_O, 0.093 (NH4)6M_O7_O_24_•4H_2_O) [43,44,45]. Twenty days after transplanting (DAT), 50 mM of NaCl was dissolved in half-strength Hoagland’s solution for irrigation, and the salt concentration was increased continually within five days to reach 150 mM at 25 DAT, which was then used for irrigating the cotton seedlings to mimic the soil salinity most cotton in northwestern China is subjected to. The soil electrical conductivity (EC) value was ~0.53 dS m^−1^ in the initial soil, it increased to 5.42 dS m^−1^ at the beginning of the 150 mM NaCl application, and it finally reached a value of 5.46 dS m^−1^ at harvesting (Table 3). During the 10 days of irrigation with 150 mM NaCl, the soil EC remained almost stable during the salt application period. The soil pH was alkaline and remained almost constant during the whole experiment (Table 3).

From 25 to 35 DAT, the seedlings were subjected daily to exogenous foliar applications of GB and SA according to the experimental design described in Table 4. The experiment was arranged using a completely randomized design performed in triplicate. At 35 DAT, the seedlings in all treatments were harvested to measure the total N content, ion concentrations, and endogenous GB and SA contents.

### 4.2. Determination of Plant Growth Parameters, pH, and EC

Plant growth parameters, including plant height, leaf area, LWP, biomass, and *PLC*, were measured three times at 5-day intervals during the 150 mM salinity stress period (20–35 DAT). Plant height was manually measured with a ruler from the surface of the soil to the apex of the leaf. The leaf area was measured using a leaf area meter (model 3050A, Li-Cor Biosciences, Lincoln, NE, USA). The LWP was measured using a WP4C, Dewpoint Potential Meter [46]. The dry masses were determined after oven-drying at 105 °C for 24 h. The *PLC* was measured using an XYL’EM-Plus. Before the measurement of *PLC*, the high-pressure (HP) reservoir was filled with 0.5 L distilled water and pressurized to 1 or 2 bars. The water valve was set to WATER to remove any existing bubbles from the HP reservoir. The low-pressure (LP) reservoir was rinsed several times with pure water to remove any potential particles, and then the stem sample was installed under LP mode to measure the *PLC* values [47]. The initial hydraulic conductance (*K*) was first measured, and the sample was flushed two times to determine the saturated hydraulic conductance (*K*′); the *PLC* was computed as:$$PLC\;=\;100\;\left.\left(1-\;\frac{K^{\prime }}{K}\right.\right)$$

The values of soil EC were measured with a pH meter (Mettler Toledo 320-S, Shanghai Bante Instrument CO., Ltd., Shanghai, China).

### 4.3. Determination of the Nutrient Content

Sampling and quantification of the total N content in plant materials containing trace amounts of nitrate were described elsewhere [48]. The extraction and determination of Na^+^ and K^+^ ions in leaves were performed according to Xu, et al. [49]. Shoots and roots were oven-dried for 48 h at 60 °C. The Na^+^ and K^+^ levels were determined with an atomic absorption spectrophotometer (spectra AA 220, Varian, Palo Alto, CA, USA). Concentrations of Ca^2+^ and Mg^2+^ were determined by inductively coupled plasma spectroscopy (ICP) using a Varian Vista-Pro CCD Simultaneous ICP-OES instrument (Victoria, Australia) [50].

### 4.4. Statistical Analysis

All experimental data were expressed as means ± standard deviation. One-way analysis of variance (ANOVA) was performed using SPSS 23.0 (IBM Corporation, New York, NY, USA). All treatment means (n = 3) were compared for any significant differences using Duncan’s multiple range tests at *p* < 0.05. The relationships between parameters were examined based on Pearson’s correlation co-efficiency method. Data fitting and graphical presentation were carried out in Origin-Pro 2017 (Origin Lab, Northampton, MA, USA). A general linear regression model was used to fit the relationships between parameters.

## 5. Conclusions

Our results demonstrated that the 150 mM NaCl regime increased Na^+^ accumulation, decreased K^+^, Ca^2+^, and Mg^2+^ uptake, and adversely affected the growth and biomass accumulation in cotton seedlings. However, foliar supplementation with 5.0 mM GB or 1.0 mM SA under NaCl stress alleviated Na^+^ toxicity and promoted nutrient uptake, along with growth biomass accumulation, in cotton seedlings. Based on our results, foliar spraying of exogenous GB at 5.0 mM, and SA at 1.0 mM was the most effective treatment for protecting cotton seedlings against NaCl-induced damage.

## Figures and Tables

**Figure 1 plants-10-00380-f001:**
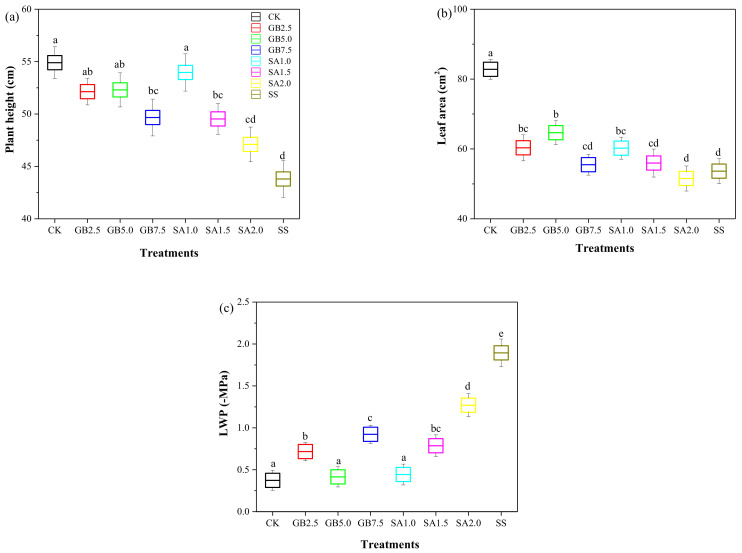
Effect of exogenous glycine betaine (GB) and salicylic acid (SA) on (**a**) plant height, (**b**) leaf area, and (**c**) leaf water potential (LWP) of 150 mM NaCl-stressed seedlings. Treatments: CK, control; GB2.5, 2.5 mM GB; GB5.0, 5.0 mM GB; GB7.5, 7.5 mM GB; SA1.0, 1.0 mM SA; SA1.5, 1.5 mM SA; SA2.0, 2.0 mM SA; SS, salt stress (150 mM) without supplementation. Values are means ± standard deviation (n = 3). Different letters represent significant differences at *p* < 0.05 between the experimental treatments.

**Figure 2 plants-10-00380-f002:**
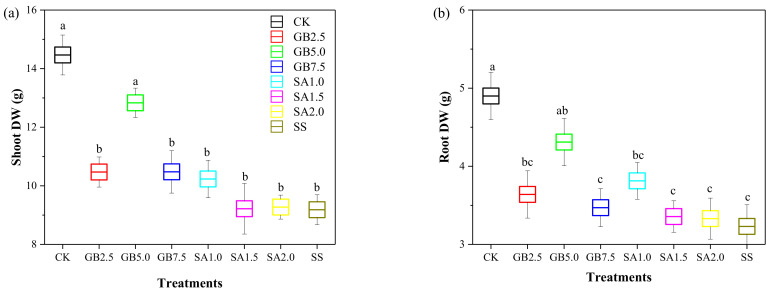
Effect of exogenous glycine betaine (GB) and salicylic acid (SA) on (**a**) shoot dry weight (DW) and (**b**) root dry weight (DW) of 150 mM NaCl-stressed seedlings. Treatments: CK, control; GB2.5, 2.5 mM GB; GB5.0, 5.0 mM GB; GB7.5, 7.5 mM GB; SA1.0, 1.0 mM SA; SA1.5, 1.5 mM SA; SA2.0, 2.0 mM SA; SS, salt stress (150 mM) without supplementation. Values are means ± standard deviation (n = 3). Different letters represent significant differences between the treatments at *p* < 0.05.

**Figure 3 plants-10-00380-f003:**
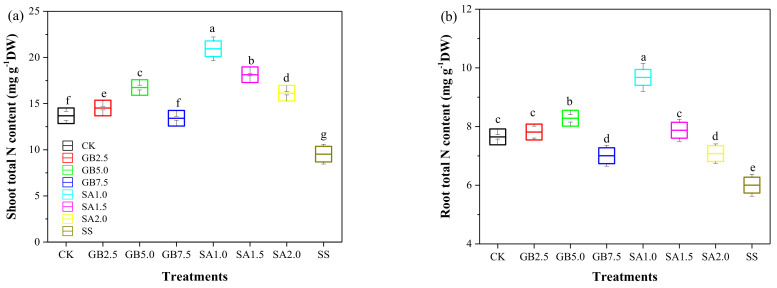
Effect of exogenous glycine betaine (GB) and salicylic acid (SA) on (**a**) shoot total nitrogen (N) content and (**b**) root total N content under the 150 mM NaCl condition. Treatments: CK, control; GB2.5, 2.5 mM GB; GB5.0, 5.0 mM GB; GB7.5, 7.5 mM GB; SA1.0, 1.0 mM SA; SA1.5, 1.5 mM SA; SA2.0, 2.0 mM SA; SS, salt stress (150 mM) without supplementation. Values are means ± standard deviation (n = 3). Different letters represent significant differences between the experimental treatments at *p* < 0.05.

**Figure 4 plants-10-00380-f004:**
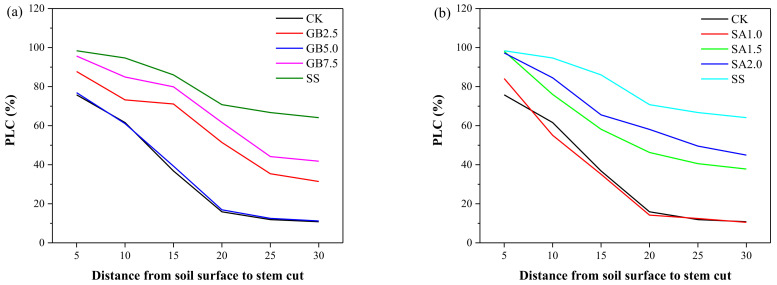
Effect of (**a**) exogenous glycine betaine (GB) and (**b**) salicylic acid (SA) on the percentage loss of conductivity (PLC) under the 150 mM NaCl regime. Treatments: CK, control; GB2.5, 2.5 mM GB; GB5.0, 5.0 mM GB; GB7.5, 7.5 mM GB; SA1.0, 1.0 mM SA; SA1.5, 1.5 mM SA; SA2.0, 2.0 mM SA; SS, salt stress (150 mM) without supplementation.

**Figure 5 plants-10-00380-f005:**
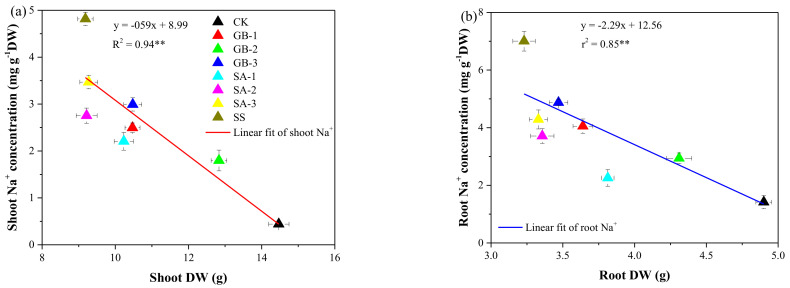
Relationship between (**a**) shoot Na^+^ concentration and shoot dry weight (DW), and (**b**) root Na^+^ concentration and root DW. Treatments: CK, control; GB2.5, 2.5 mM GB; GB5.0, 5.0 mM GB; GB7.5, 7.5 mM GB; SA1.0, 1.0 mM SA; SA1.5, 1.5 mM SA; SA2.0, 2.0 mM SA; SS, salt stress (150 mM) without supplementation. Values are means (n = 3), and error bars represent standard deviations; * and ** indicate significance levels of *p* < 0.05 and *p* < 0.01, respectively.

**Figure 6 plants-10-00380-f006:**
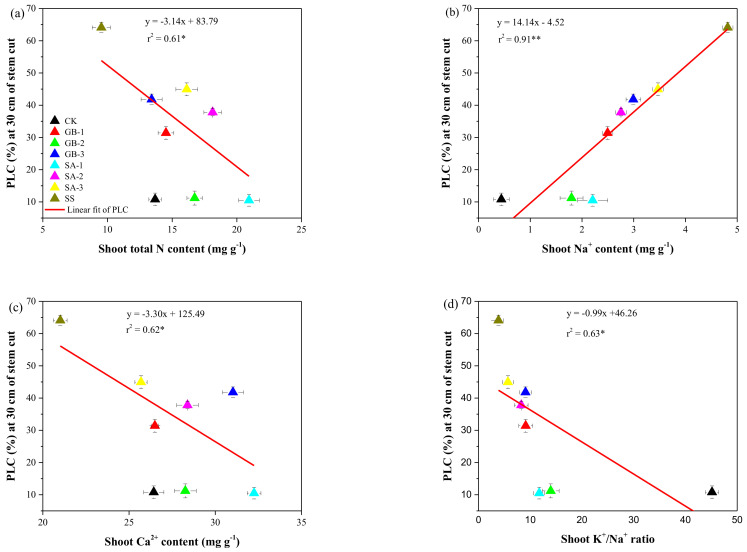
Relationship between (**a**) percentage loss of conductivity (PLC) and shoot total nitrogen (N) content, (**b**) PLC and shoot Na^+^ content, (**c**) PLC and shoot Ca2+ content, and (**d**) PLC and K^+^/Na^+^ ratio. Treatments: CK, control; GB2.5, 2.5 mM GB; GB5.0, 5.0 mM GB; GB7.5, 7.5 mM GB; SA1.0, 1.0 mM SA; SA1.5, 1.5 mM SA; SA2.0, 2.0 mM SA; SS, salt stress (150 mM) without supplementation. Values are means (n = 3), and error bars represent standard deviations; * and ** show significance levels at *p* < 0.05 and *p* < 0.01, respectively.

**Table 1 plants-10-00380-t001:** Effect of exogenous glycine betaine (GB) and salicylic acid (SA) on shoot ion concentrations under the 150 mM NaCl regime.

Treatments	Shoot Ion Concentrations (mg × g^−1^ Dry Weight)				Ratios	
	K^+^	Na^+^	Ca^2+^	Mg^2+^	K^+^/Na^+^	Ca^2+^/Mg^2+^
CK	19.91 ± 0.28 e	0.44 ± 0.02 g	26.43 ± 0.59 d	3.86 ± 0.40 b,c	45.13 ± 1.25 a	6.88 ± 0.56 a,b
GB2.5	22.73 ± 0.12 d	2.49 ± 0.01 d	26.49 ± 022 d	3.80 ± 0.43 b,c	9.11 ± 0.32 d	7.88 ± 1.13 a
GB5.0	25.08 ± 0.43 c	1.79 0.22 f	28.26 ± 0.63 c	4.07 ± 0.02 b	13.95 ± 0.62 b	6.93 ± 0.18 a,b
GB7.5	27.19 ± 0.21 a	2.99 ± 0.04 c	31.02 ± 0.61 b	4.99 ± 0.03 a	9.08 ± 0.12 d	6.21 ± 0.09 b
SA1.0	25.88 ± 0.88 b	2.20 ± 0.29 e	32.25 ± 0.40 a	5.16 ± 0.40 a	11.73 ± 0.06 c	6.26 ± 0.41 b
SA1.5	22.74 ± 0.13 d	2.75 ± 0.11 c	28.39 ± 0.63 c	4.04 ± 0.01 b	8.26 ± 0.31 d	7.02 ± 0.14 a,b
SA2.0	19.74 ± 0.01 e	3.46 ± 0.01 b	25.69 ± 0.15 d	3.81 ± 0.08 b,c	5.69 ± 0.09 e	6.73 ± 0.10 b
SS	18.59 ± 0.12 f	4.81 ± 0.10 a	21.01 ± 0.39 e	2.94 ± 0.50 d	3.86 ± 0.02 f	7.27 ± 1.23 a,b

Note: Values are means ± standard deviation (n = 3). Abbreviations: CK, control; GB2.5, 2.5 mM GB; GB5.0, 5.0 mM GB; GB7.5, 7.5 mM GB; SA1.0, 1.0 mM SA; SA1.5, 1.5 mM SA; SA2.0, 2.0 mM SA; SS, salt stress (150 mM) without supplementation. Different letters represent significant differences between the experimental treatments at *p* < 0.05.

**Table 2 plants-10-00380-t002:** Effect of exogenous glycine betaine (GB) and salicylic acid (SA) on root ion concentrations under the 150 mM NaCl regime.

Treatments	Root Ions Concentrations (mg × g^−1^ Dry Weight)				Ratios	
	K^+^	Na^+^	Ca^2+^	Mg^2+^	K^+^/Na^+^	Ca^2+^/Mg^2+^
CK	17.64 ± 0.55 c	1.41 ± 0.03 g	7.21 ± 0.21 d	3.85 ± 0.13 a,b	12.43 ± 0.92 a	1.87 ± 0.11 e
GB2.5	18.39 ± 0.24 b	4.05 ± 0.05 c,d	7.67 ± 0.69 d	2.24 ± 0.01 d	4.53 ± 0.32 d	3.42 ± 0.31 b
GB5.0	18.75 ± 0.26 b	2.94 ± 0.58 e	8.22 ± 0.59 c	3.12 ± 0.01 c	6.37 ± 0.61 c	2.63 ± 0.18 c,d
GB7.5	21.72 ± 0.18 a	4.87 ± 0.11 b	8.83 ± 1.01 b	3.54 ± 0.11 b	4.45 ± 0.21 d	2.49 ± 0.20 c,d
SA1.0	21.70 ± 0.63 a	2.25 ± 0.29 f	9.41 ± 0.10 a	4.13 ± 0.34 a	9.63 ± 0.64 b	2.29 ± 0.17 d
SA1.5	18.51±0.27 b	3.71±0.25 d	8.88±0.24 c	3.62±0.50 b	4.99±0.86 d	2.48±0.27 c,d
SA2.0	17.56±0.56 c	4.28±0.33 c	8.29±0.29 d	3.11±0.02 c	4.09±0.45 d	2.66±0.09 c
SS	16.77±0.18 d	7.00±0.34 a	6.84±0.28 e	1.80±0.15 e	2.39±0.11 e	2.79±0.15 a

Note: Values are means ± standard deviation (n = 3). Abbreviations: CK, control; GB2.5, 2.5 mM GB; GB5.0, 5.0 mM GB; GB7.5, 7.5 mM GB; SA1.0, 1.0 mM SA; SA1.5, 1.5 mM SA; SA2.0, 2.0 mM SA; SS, salt stress (150 mM) without supplementation. Different letters represent significant differences between the experimental treatments at *p* < 0.05.

**Table 3 plants-10-00380-t003:** Soil pH and EC during the experiment.

Soil Conditions	pH	EC (dS m^−1^)
Initial soil	8.12	0.53
At 150 mM NaCl application	8.18	5.42
After harvesting	8.56	5.46

Note: Data for pH and electrical conductivity (EC) are means of all saline treatments at different time points.

**Table 4 plants-10-00380-t004:** The detailed experimental treatments.

Treatment Label	NaCl Dose (mM)	GB Dose (mM)	SA Dose (mM)
CK	-	-	-
GB2.5	150	2.5	-
GB5.0	150	5.0	-
GB7.5	150	7.5	-
SA1.0	150	-	1.0
SA1.5	150	-	1.5
SA2.0	150	-	2.0
SS	150	-	-

Abbreviations: CK, control; GB2.5, 2.5 mM GB; GB5.0, 5.0 mM GB; GB7.5, 7.5 mM GB; SA1.0, 1.0 mM SA; SA1.5, 1.5 mM SA; SA2.0, 2.0 mM SA; SS, salt stress (150 mM) without supplementation.

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
