# Peer review of "Application of Exogenous Protectants Mitigates Salt-Induced Na+ Toxicity and Sustains Cotton (Gossypium hirsutum L.) Seedling Growth: Comparison of Glycine Betaine and Salicylic Acid"

_plants, 2021, doi:10.3390/plants10020380_

Round 1

Reviewer 1 Report

The authors have made changes to the previous version of the manuscript, improving its readability. The References section must be modified as it is not possible to compare the references cited in the text. 

Author Response

Response letter to reviewer 1

Note: The line number mention by editor in the comments are kept same which is according to the manuscript, whereas the line number mention in the response for the corresponding comments line is the new line number of modified version of manuscript.

Comment:

The authors have made changes to the previous version of the manuscript, improving its readability. The References section must be modified as it is not possible to compare the references cited in the text. 

Response:

Thank you for your appreciation. We have numbered the list of references as suggested (Lines 532-651).

Reviewer 2 Report

The authors have provided adequate responses to most of my previous concerns. The revised version of the manuscript has improved with the changes made, but there are still a couple of issues that should be addressed before considering the acceptance of the work.

My main concern remains with the discussion of the results obtained:

a) Lines 381-382: The entire paragraph refers to results that have been removed in the present version of the manuscript.

b) Although it has been improved, the discussion of the results remains superficial. In most of the text the discussion does not go beyond repeating the description of the results and briefly citing previous studies in other species. The authors should try to fit their observations in a more robust framework, where the effect of the treatments carried out should be integrated as a whole and not simply examine the effect in each parameter measured separately. On the other hand, the authors should try to discuss in more depth what may be the possible causes of the protective effects of GB or SA against saline stress in the plant (and in cotton in particular). This would help a future reader of this work to obtain an intellectual benefit that goes beyond the verification that GB or SA treatments work in cotton subjected to saline stress, as suggested by previous studies in other plants

Another comments:

c) The methodology used to carry out the correlation analyses should be better detailed in the materials and methods section.

d) Also, the list of references is not numbered!

Author Response

Response letter to reviewer 2

Note: The line number mention by editor in the comments are kept same which is according to the manuscript, whereas the line number mention in the response for the corresponding comments line is the new line number of modified version of manuscript.

Comments and Suggestions for Authors

The authors have provided adequate responses to most of my previous concerns. The revised version of the manuscript has improved with the changes made, but there are still a couple of issues that should be addressed before considering the acceptance of the work.

My main concern remains with the discussion of the results obtained:

Comment a:

  1. a) Lines 381-382: The entire paragraph refers to results that have been removed in the present version of the manuscript.

Response a:

The paragraph has been revised and is now coherent to our results. We added the reference of our previous article (Lines 393-407).

Comment b:

  1. b) Although it has been improved, the discussion of the results remains superficial. In most of the text the discussion does not go beyond repeating the description of the results and briefly citing previous studies in other species. The authors should try to fit their observations in a more robust framework, where the effect of the treatments carried out should be integrated as a whole and not simply examine the effect in each parameter measured separately. On the other hand, the authors should try to discuss in more depth what may be the possible causes of the protective effects of GB or SA against saline stress in the plant (and in cotton in particular). This would help a future reader of this work to obtain an intellectual benefit that goes beyond the verification that GB or SA treatments work in cotton subjected to saline stress, as suggested by previous studies in other plants

Response b:

We fully agree with your observation. Thus, we have improved the discussion of our result with a general concept about the effect of exogenous GB and SA under salt stress. It also increased the number of references in our manuscript, and those references could be useful for readers to understand more aspects about exogenous application of GB and SA under saline conditions. We’re hopeful the improvement we made could satisfy you as our discussion was already long (Lines 269-269 and 413-417).

Comment c:

  1. c) The methodology used to carry out the correlation analyses should be better detailed in the materials and methods section.

Response b:

The methodology used to carry out the correlation analyses is clearly detailed in the materials and methods section (Lines 506-510).

Comment d:

  1. d) Also, the list of references is not numbered!

Response b:

We have numbered the list of references as suggested (Lines 532-651).

Round 2

Reviewer 2 Report

WIth the last changes made by the authors, the manuscript is now suitable to be published in Plants

This manuscript is a resubmission of an earlier submission. The following is a list of the peer review reports and author responses from that submission.

Round 1

Reviewer 1 Report

Manuscript entitled “Application of exogenous protectant mitigate salt-2 induced Na+ toxicity and sustain cotton (Gossipium 3 hirsutum L.) seedling growth: comparison of glycine 4 betaine and salicylic acid” by Kader et al describes the study of the effects of foliar application of Glycine betaine (GB) or Salycilic Acid (SA)  in cotton plants exposed to saline stress conditions. The authors assay three concentrations for each molecule, comparing with unstressed plants (control) or stressed plants without any foliage treatment. The authors follow the plant stress response by determining several parameters, including growth-related (height, leaf area, dry weight), nutrition-related (N content, Na, K, Ca and Mg) as well water-related (leaf water potential and stem hydraulic conductivity).

The proposed approaches and the treatment strategy used seem both correct for the most part.  However, the way in which the authors handle, show and interpret the results obtained is highly debatable (see my comments below). On the other hand, the discussion of the results obtained needs a very considerable improvement, but, among other aspects, in many cases it does not go beyond a repetition of the description of the results (again, see my comments below).

For all these reasons, I consider that the present manuscript is still far from enough, in my opinion, to consider its publication in this journal.

Some comments and suggestions about your paper:

a)       Regarding introduction:

Lines 14-15:  please check English grammar in the sentence “To deal with the noxious impacts of salinity stress is becoming one of the main goals of agricultural researchers worldwide.”

Lines 35-36 Check English grammar in the sentence” The salt accumulations of soil are often the result of watering with sodium chloride (NaCl) contains in irrigation water”. Perhaps is better modify with “The salt accumulation of soil is often the result of irrigation with water containing sodium chloride (NaCl)”

Lines 41-43: If the sentence “Therefore, it is better to improve cotton growth performances and its adaptation to saline conditions by exogenous foliar supplementations with GB and SA.” refers to previous studies, it needs a reference.  I the authors are referring to the conclusions of the work presented, I think that is premature to include the sentence in the introduction.

Line 44:” Na+, Cl-, Mg2+ and Ca2+ ions contribute in soil salinization”. The authors need to clarify better what is this contribution for each cation/anion.

Lines 44-45 “Approximately 20% of farmland and 50% of cropland is saline worldwide” this sentence need to be rewritten to a more understandable version. Whats means here “saline cropland”?

Line 56:  please replace “an ion homeostasis” with “and ion homeostasis”

Line 59: “Salicylic acid (SA) is an important phytohormone responding to plant growth “Phytohormones are chemical signals that are used by the plants in their growth and development as well as in their interaction with the environment. If the levels of a hormone experiment changes during a process, this implies that this hormone may have a role in the signaling mechanisms of that process, but not that it "responds" to that process.

Line 62:  which kind of nutrients increase its levels? Mineral nutrients?

 Line 63: “Plants growth is subjected to salinity stress” Literally this sentence could give to the reader the impression that there is a requirement of salinity stress to plant growth, please rephrase.

Line 64-65: Check English grammar in the sentence “The amount of water translocated from roots to shoot defined the content of substances transferred to the shoot”

Line 71-72: “Under saline conditions, positives impacts were observed on growth and yield of crop with 71 exogenously sprayed GB and SA [4]. This reference only mentions works that involve GB and not SA.

Lines 76-79: The authors must include in the introduction previous observations that justifies the predictions listed in the hypothesis formulated in these lines. Without these previous references, this hypothesis sounds more like a summary of the results obtained. Also, please correct “loos” with “loss” in line 78

b)      Regarding Results:

- Lines 87-89: “The medium dose (5 mM) of exogenous GB and the lowest dose (1 mM) of exogenous SA, showed best performances in enhancing plant growth parameters under salinity stress”. This sentence is not accurate. Since you do not have statistical differences in plant height of leaf area between 2.5 mM and 5 mM GB treatments, neither in the case of the leaf area between treatments 1 mM and 1.5 mM SA (figures 1a, 1b)

-lines 89-91: “The above plant growth indicators have insignificantly responded to the exogenous foliar supplementation with the highest foliar sprayed SA under saline condition.” Again, this description is not accurate, because you have a significant difference between the LWP in plants treated with 2 mM SA compared untreated- salt stressed plants (SS). Figure 1c. Also I am a little surprised that in Figure 1c some treatments with clear overlapping between their error bars are considered with significant differences( like between GB-1 and GB-2 treatments, or between SA-1 and SA-2)

Lines 101-102: “The biomass of seedling in both roots and shoot was significantly increased by the treatment with 5 mM exogenous GB under NaCl regime”. Again, the description is not accurate, because there are not statistical differences in root DW between 2,5 and 5 mM GB treatments (figure 2B).

Lines 112-113: “The total N content in both shoot and roots were similarly positively affected by the two osmolytes treatments under high NaCl  regime, where the highest total N accumulation was measured in the treatment of 1 mM exogenous SA”. I do not think that SA could be classified as an osmolyte. Again, I am very surprised that several treatments are considered with significant differences looking the overlapping of the error bars between GB  or SA treatments, and sometimes (GB-3, in both shoot and root or SA3 in roots) there is also an overlapping with the error bars of SS treatment in both figures 3a and 3b.

Line 124: PLC or PCL?

Figure 4: - Why SS treatment is not included in figure 4a? Why CK treatment is not included in figure 4b?

Line 140: please replace ”in contrary” with “on the contrary”

Lines 163-164: “Overall exogenous foliar GB and SA treatments, the medium dose of GB showed the best performance in mitigating Na+ toxicity” Again, the sentence is not completely accusate. Since GB1 and GB2 treatments have not significant differences in K+ levels on roots.

Most of the correlations shown in Figures 5 and 6 are highly debatable.

Line 198: Again, SA is a phytohormone. It is not an endogenous osmolyte.

Figure 7: Again, I am surprised that some treatments appear with significant differences although in the graph it is evident that their error bars overlap (figure 7a, GB-3 compared with SA-1 or GB-1 and GB-2 or SA- 1and SA-2)

Lines 201-203: “Exogenous foliar application of all GB and SA levels, except for the highest level of SA, significantly increased the endogenous GB content in seedling leaves under NaCl stress,” The sentence is again inaccurate, because GB levels differences are not significant for GB-1 treatment compared to levels in SS plants.

c)       Regarding Discussion

Lines 286 and 290? What is the meaning of the expressions “full of N nitrification” or “full of N nutrition capacities”?

Lines 286-288: “All levels of foliar sprayed GB and SA significantly improved the total N content in both root and shoot compared to the NaCl-stressed 287 treatment alone (Figure 3)”  I have to disagree again with this sentence, at least looking how error bars overlap in the cases of GB-3, SA-3 and SS treatments.

Line 291-293: “This ability of exogenous GB and SA to enhanced the total N concentration in cotton plants subjected to severe NaCl stress is caused by their molecular structures and the convenient doses application”. The authors give at this point a very superficial explanation/interpretation that need to be better developed or extended. In which way the molecular structures of these two molecules are causing this increase in N content? Why a “convenient” dosage has an influence? Also, the expression “to enhanced” should be changed with “to enhance”

Lines 293-294: “Recently it was reported in a review that GB is one of the compatible osmolytes containing a high rate of N, as found in Poaceae”. If the authors suggests that the increase in N content of the GB-treated plants is being caused by the incorporation of the sprayed GB to the nitrogen metabolism of the plant, then the plants treated with the GB-3 dose should have the higher N content, and this do not happen…

Lines 283-299: What is the justification for jointly discussing the incorporation of nitrogen and the PLC of the leaves? On the other hand, the discussion of the obtained PLC data is limited to a new repetition of the description of the obtained data, without any integration with previous data. In addition, there is not any mention to the correlations described in figure 6

Line 307-312: Again, the discussion of the final part of the work does not go beyond a mere repetition of the description of the results. In fact, they only mention one study that contradicts their observation: “In agreement with this, exogenously sprayed SA was found to ameliorate the endogenous accumulation of GB in Vigna radiate subjected to salinity stress”. The authors observe the opposite result ... the treatment with SA increases the levels of GB in the treatments SA-1 and SA-2!

Also, I think the authors overlook a couple of interesting details regarding the levels of GB and SA obtained in each treatment:

-GB treatment prevents the rise in SA levels. I would suggest additional experiments to the authors to explore whether this absence of rise is due to less stress or direct blockage of the synthesis. It would be convenient to treat CK plants with GB and measure SA levels.

-the protective effects of SA treatment may be mediated, in part, by the increases in GB levels seen in SA-1 and SA-2. It would also be interesting to see if this induction in the production of GB would also be observed when treating CK plants with SA.

Lines 382-384: “Based on our results, the exogenous foliar sprayed SA with 382 its lowest concentration seems to be more credible than exogenous GB in protecting cotton plants from the NaCl-induced damages.”  I have not been able to locate any part of the work where the authors perform a comparison or a discussion to suggest which of the two external treatments are more suitable to mitigate the saline stress damage.

d)      Regarding Figures:

-Figure legends should include all the necessary information to inform the reader about the experiment performed without having to resort to the main text. Hence, the information about NaCl, GB and SA treatments should be included.

e)      Regarding References: 

Reference format needs revision. The list of references is not numbered following the order of this first appearance in the text.  An example: from number [3] in line 38 the authors jump to number [7] in line 41, while references 4 and 5 appear for the firt time in line 54, why?

Reviewer 2 Report

The work reports many data relating to different foliar treatments with solutions with different concentrations of glycine betaine (GB) and salicylic acid (SA) on cotton subjected to high NaCl stress.

In the Introduction section I suggested to move the sentence of lines 39-43 "Cotton as largest fiber source ... exogenous foliar supplementations with GB and SA" to line 69.

In the Materials and methods section, the analysis methods used to determine the parameters are not clearly reported, in detail:

the reference used for the quantification of glycine betaine (GB) (reference 42) seems out of date. GlyBet was converted to its n ‐ butyl ester and determined by fast atom bombardment ‐ MS according to Rhodes et (1987).

Please indicate the instruments and conditions employed for GB and SA quantitation.

The methods used to quantify nitrogen, sodium, potassium, calcium and magnesium should also be described. The description given bibliographic references indicated does not seem clear to me.

check that the chemical formulas are spelled correctly

In the results and discussion section I would add a paragraph explaining the correlation between leaf treatment and the various parameters analyzed

Reviewer 3 Report

Dears Authors: Mounkaila H.A.K et al.

In my opinion the Article can be published in the Scientific Journal - Plants, after minor revision. Corrections are connected with minor methodological errors and text editing.

In my opinion, the biggest problem which I see in materials and methods chapter is connected with number of experiment series. How many series of experiments the Authors done? Please see the enclosed file. 

Enclosed please find the review of the above communication. Please carefully review all criticisms and suggestions raised by the refere.

Reviewer 4 Report

In  this paper the authors describe the effect of exogenous glycinebetaine and SA in protecting cotton seedlings fromthe negative effects of  salt stress. They found that 5 mM GB and 1 mm SA  were more effective to mitigate Na+ toxicity and to enhance biomass accumulation under high NaCl regime(150mM).  Moreover, they found that SA significantly increased both endogenous GB and SA concentrations, whereas  exogenous GB only increased endogenous GB concentration.

I find the paper clearly written even if moderate english changes are required.

Nevertheless I find the results merely descriptive and no new insight are present in the paper.

In details:

- In my opinion the salt stress applied is too hard to mimic  the salinity of a typical environment. A saline soil is generally defined as one in which the electrical conductivity (EC) of the saturation extract (ECe) in the root zone exceeds 4 dS m−1 (approximately 40 mM NaCl) at 25 °C (Munns, 2005; Jamil et al, 2011; Dezhi Wu, Guoping Zhang, 2016).

- As the authors describe strong differences among treated and non treated plants under salinity it would have been useful to show some pictures of the plants. Since the differences described are very significative a reduced growth should be appreciate in non treated plants.

- the discussion is only confirmative of results reported by other authors on other species. There are no indication of the mechanism involved in SA or GB protection on the root system or plant.

- the bibliography is poorly updated.

Reviewer 5 Report

This manuscript displays very interesting researches, but I have some questions:

Introduction:

It should be more emphasized what the novelty of this work is

Material and method:

Why the used  concentrations of GB and SA  were chosen?

Results:

Explanations under figures and tables should be more accurate - should contain GB and SA concentrations - so that they are clear to the reader without resorting to methodology.

Fig 4 - Both positive (CK) and negative control (SS) should be placed on Fig 4a and 4b

Formatting should be checked in the whole text - use superscript where necessary, e.g. Line 319, 328, 329